# Influence of Nanoparticles and Metal Vapors on the Color of Laboratory and Atmospheric Discharges

**DOI:** 10.3390/nano12040652

**Published:** 2022-02-15

**Authors:** Victor Tarasenko, Nikita Vinogradov, Dmitry Beloplotov, Alexander Burachenko, Mikhail Lomaev, Dmitry Sorokin

**Affiliations:** 1Institute of High-Current Electronics SB RAS, 634055 Tomsk, Russia; vinikitavin@mail.ru (N.V.); rff.qep.bdim@gmail.com (D.B.); bag@loi.hcei.tsc.ru (A.B.); lomaev@loi.hcei.tsc.ru (M.L.); sdma-70@loi.hcei.tsc.ru (D.S.); 2Department of Quantum Electronics and Photonics, National Research Tomsk State University, 634050 Tomsk, Russia

**Keywords:** mini-jets, diffuse discharge, spark discharge, red sprites, blue jets, ghosts

## Abstract

Currently, electrical discharges occurring at altitudes of tens to hundreds of kilometers from the Earth’s surface attract considerable attention from researchers from all over the world. A significant number of (nano)particles coming from outer space burn up at these altitudes. As a result, vapors of various substances, including metals, are formed at different altitudes. This paper deals with the influence of vapors and particles released from metal electrodes on the color and shape of pulse-periodic discharge in air, nitrogen, argon, and hydrogen. It presents the results of experimental studies. The discharge was implemented under an inhomogeneous electric field and was accompanied by the generation of runaway electrons and the formation of mini-jets. It was established that regardless of the voltage pulse polarity, the electrode material significantly affects the color of spherical- and cylindrical-shaped mini jets formed when bright spots appear on electrodes. Similar jets are observed when the discharge is transformed into a spark. It was shown that the color of the plasma of mini-jets is similar to that of atmospheric discharges (red sprites, blue jets, and ghosts) at altitudes of dozens of kilometers and differs from the color of plasma of pulsed diffuse discharges in air and nitrogen at the same pressure. It was revealed that to observe the red, blue and green mini-jets, it is necessary to use aluminum, iron, and copper electrodes, respectively.

## 1. Introduction

At present, nanopowders are widely used in various fields. Micro- and nanoparticles are used in cases of exposure to solids and liquids [1,2,3], as well as in biology [4], agriculture [5], and other fields [6]. These highly demanded areas have already become traditional; therefore, scientific teams from different countries continue to conduct intensive research aimed at studying the properties of such powders (see, for example, [7,8,9,10]). It is known that metal vapors and various particles affect the optical properties of gas-discharge plasma. In [11,12], it was demonstrated that the plasma of a repetitively pulsed spark discharge containing metal vapors can be used as a source of spontaneous radiation in the UV region of the spectrum. On the other hand, it is of interest to establish the degree of influence of micro- and nanoparticles, as well as metal vapors, on the properties of discharges in the upper layers of the atmosphere of our planet. It is known that in the upper layers of the atmosphere, at an altitude of tens of kilometers, a large number of micrometeorites burn down and evaporate [13]. As a result, vapors of different materials, including metals, appear at different altitudes. It is of interest to determine whether vapors of meteorites burning in the Earth’s atmosphere affect the properties of high-altitude discharges.

High-altitude discharges in the Earth’s atmosphere are studied by many scientific groups (see [14,15,16,17,18,19]). In recent years, the improvement of instruments for detecting various types of radiation, as well as means for capturing images from the International Space Station (ISS), have contributed to obtaining new results [19,20]. Recently, a large number of color photographs of high-altitude transient luminous events (TLEs; red sprites, elves, ghosts, blue jets, and their analogues (starters, giant jets) are referred to as TLEs) have appeared [14,15,16,17,18,19,20,21]. Figure 1 demonstrates a collage of images of TLEs, composed from photographs obtained using various sources [21].

One issue requiring more research is the determination of the color source for each of these phenomena. It is believed that the color of red sprites that initiate at altitudes of about 70 km and are visible at altitudes of 40 to 90 km is determined by the radiation of the first positive 1P system of a nitrogen N_2_ molecule [15,17,18]. This fact is also confirmed by spectral studies of sprite emission [22]. The color of blue jets, which originate from the tops of clouds at about 18 km and reach an altitude of 50 km or more, according to [15,17,19,23], is associated with the radiation of the second positive 2P system of a nitrogen molecule N_2_ and the first negative 1N system of a nitrogen N_2_^+^ ion. There is no information in the available sources about the nature of the green color of phenomena such as ghosts. However, the color of TLEs of the same type, described in different sources, may differ significantly. Thus, the elve in Figure 1 has a red color [21], while on the video taken from the ISS, it is blue-pink [20].

In laboratory experiments, with decreasing air pressure, the color of the plasma of pulsed discharges usually changes from blue to pink, while the color of the sprites observed in natural conditions is bright red, and that of jets is blue [14,15,16,17,18,19,20,21,23]. However, both sprites and jets (or their parts) can be observed at the same altitude (see [16]). Consequently, lines and bands of radiation of a different nature can determine the color of observed TLEs. On the other hand, it was shown in [24,25] that in diffuse and spark discharges in air, mini-jets are observed near electrodes made of aluminum, stainless steel, and copper, the colors of which are red, blue, and green, respectively.

The aim of this work is to study the optical properties of plasma of diffuse and spark discharges in an inhomogeneous electric field at different pressures of air, argon, nitrogen, and hydrogen with the injection of electrode material into the discharge region due to the explosion of microprotrusions on the electrode surface and/or transition to a spark. The color of mini-jets observed in such discharges is compared with that of high-altitude atmospheric discharges (blue jets, red sprites, and ghosts).

## 2. Experimental Setup and Methods

The study of the discharge modes and optical characteristics of the formed plasma was carried out using electrodes of various shapes, made of different materials. A cone-shaped cathode and a plane anode (“point-plane” gap geometry) were used in most experiments. This gap geometry provides the formation of diffuse discharges in various gases at a relatively low amplitude of voltage pulse due to the strongly non-uniform electric field strength distributions and the generation of runaway electrons [26,27]. It is known that high-altitude discharges are also accompanied by the generation of runaway electrons and other high-energy particles [23].

A sketch of an experimental setup, consisting of a voltage-pulsed generator (NPG-15/2000N or NPG-18/3500N, generators differ in the ranges of adjustment of the amplitude and repetition rate of voltage pulses), a 3 m long coaxial cable with a wave impedance of 75 Ω, and a discharge chamber, is presented in Figure 2.

The NPG generators form voltage pulses *U*_g_ with amplitudes ranging from 12 to 18 kV (the voltage across the discharge gap doubled due to its reflection from a “cold” gap) with a full width at half maximum (FWHM) and a rise time of ≈6 and ≈3 ns, respectively. The studies were carried out at a pulse repetition rate *f* from 60 to 1000 Hz. Some experiments were performed in single-pulse mode. In addition, a RADAN-220 generator with a voltage pulse amplitude in an incident wave *U*_g_ of 120 kV and a duration at a matched load of 2 ns was used.

Aluminum, copper, stainless steel, and tungsten were used as electrode materials. The high-voltage, cone-shaped electrode (2 in Figure 2) was made of D14 aluminum, copper, or stainless steel. It had an apex angle of 40 degrees. However, the radius of its tip rounding varied from 70 to 380 μm in the experiments. In several experiments, the cone-shaped electrode was replaced by a bundle of 0.2-mm-diameter tungsten wires with sharp ends. The plane electrode (1 in Figure 2) was made of aluminum, stainless steel, or copper. A gas (air, nitrogen, argon, and hydrogen) pressure varied from 0.1 to 760 Torr. The gap width varied from 1 to 12 mm.

To increase the concentration of the sputtered and evaporated electrode material, as well as the intensity of the discharge plasma radiation, the experiments were carried out under the ignition of the discharge in a repetitively pulsed mode at a voltage of tens of kilovolts. In some experiments, the gas pressure corresponded to that of air at altitudes where blue jets and red sprites are observed.

The use of an electrode with a small radius of curvature as a high-voltage cathode made it possible to determine the conditions under which the generation of runaway electrons takes place. When registering the runaway electron beam current, a plane anode made of a thin aluminum foil or a grid with a light transmission of 67% was used. The results of studies of the properties of a beam of runaway electrons generated in a repetitively pulsed discharge ignition mode on a similar experimental setup are described in detail in [26].

The discharge glow images were photographed with a Sony A100 digital camera (Kuala Lumpur, Malaysia). The colors of red sprites, blue jets, and other TLEs in the images given in [17,18,19,23] and [20,21] were compared with the color of the repetitively pulsed discharge plasma.

Emission spectra from the different discharge zones were recorded with an EPP-2000C spectrometer (StellarNet Inc. (Tampa, Fl, USA), λ = 192–854.5 nm). Waveforms of the voltage from a capacitive voltage divider and electron beam current from a collector were recorded with a Tektronix MDO 3104 oscilloscope (1 GHz, sampling rate 5 GS/s).

## 3. Results

### 3.1. Pulse Breakdown Conditions for Laboratory and Atmospheric Discharges

It is well known that the gap width, the shape and material of the electrodes, the amplitude and rise time of the voltage pulse, the gas pressure and gas type, as well as the method of the gas pre-ionization, determine the breakdown characteristics. When studying laboratory discharges, experimental conditions are relatively easy to control, including measuring the current of runaway electrons [25,26]. In addition, the amplitude and duration of a voltage pulse, the current density, and the energy stored in a high-voltage generator can be varied over a wide range.

The situation with discharges in the upper layers of the Earth’s atmosphere is much more complicated. In the case of TLEs, it is necessary to determine not only the altitude of their formation and air composition in this region, but also the electrical properties of the atmosphere (electric field strength, discharge current density, and pulse duration). However, high-altitude atmospheric discharges are transient and electrical parameters are very difficult to measure. The initiation of TLEs depends on the air temperature, as well as the height of the clouds, their density, and the distribution of charges in them. In addition, the formation of these discharges is associated with the appearance of lightning in the lower layers of the Earth’s atmosphere. High-energy particles from space, from the Sun, and generated in strong electric fields, are important for charge accumulation and discharge initiation. VUV/UV radiation from the Sun plays an important role in the ionization of the Earth’s atmosphere. All these factors can affect the type of discharge and its color. In this paper, the main focus is on the study of the influence of metal vapors and particles appearing in the gap on the coloration of various discharge regions. Data on the appearance of micro- and nanoparticles in the gap, as well as metal vapors, that affect the color of the discharge plasma, are given below.

### 3.2. Formation of Micro- and Nanoparticles during Spark and Diffuse Discharges

It should be noted that the formation of particles and electrode material evaporation have previously been studied in detail for vacuum discharges (see, for example, [28,29,30]). The presence of particles in those experiments was observed under various discharge modes. Double diffuse jets (streamers) from the ends of particle tracks were detected for the first time. Below are photographs showing the appearance of particles in the discharge. The cone-shaped, high-voltage electrode used in most of the experiments is placed at the bottom of the images. In the case of electrodes that are poorly distinguishable against the background of the glow of the discharge plasma, their outlines are shown by white lines.

First, the appearance of particles in a spark discharge is demonstrated. The photographs in Figure 3 show the spark discharge glow, the cone-shaped electrode with a small radius of the tip rounding, luminous tracks of particles, and diffuse jets at the ends of the tracks.

The photograph was captured in the single-pulse mode of the spark discharge in the 1 mm long gap filled with atmospheric-pressure air. A short gap was used to increase the current density during the spark stage of the discharge. The spark phase of discharge combustion provides the greatest contribution to the intensity of radiation from the discharge gap. Figure 3a demonstrates a bright spot (the glow from the spark channel), which touches the tip of the cone-shaped electrode with its bottom part and adjoins the flat electrode with its top part. However, even with intense radiation from the spark channel, the tracks of two microparticles are visible. They were emitted from the point of contact between the spark and the cone-shaped electrode. Under these conditions, the brightness of the tracks increases with distance from the electrode. The particle trajectories are different: one of them (2) abruptly changes the direction of its motion.

Figure 3b,c shows zoomed images of the particles’ tracks, which, as already noted, can change direction and end in jets. The smooth change in the direction of movement of the microparticles can be explained by the influence of an electric field in this direction. The nature of the glow of the particles in Figure 3 corresponds to the glow of a micrometeorite that burns down in the Earth’s atmosphere [13]. The brightness of the micrometeorite (the particle) glow increases towards the track’s end. This cannot be explained by an increase in the particle velocity, since the particle stops. This occurs after the voltage pulse action. We believe that the increase in the radiation intensity of the particle is due to its heating during deceleration on gas particles.

Let us note an important feature in the formation of tracks, shown in Figure 3. At the ends of the tracks, jets (streamers) are visible. Usually, they are oriented in opposite directions. The appearance of these diffuse jets at the ends of the tracks is due to the formation of streamers from the plasma surrounding the particle surface. As is known, streamers are formed when the threshold plasma concentration for the electric field at a given point is reached. The formation of streamers is confirmed by the formation of two rectilinearly propagating jets from the same region (3), as well as an abrupt change in the direction of propagation of the upper jet (4) (see Figure 3c). Streamers appear to carry away a significant part of the charge, which leads to a deceleration of the microparticle. In this case, the discharge was formed at the relatively high air pressure (760 Torr) and the small interelectrode distance, which made it possible to observe tracks of individual particles. The observation of microparticle tracks is facilitated in the case of the ignition of a spark discharge with the high current density on the electrodes.

With decreasing gas pressure and increasing interelectrode distance, the current density decreased; sparks do not have time to form at short voltage pulse durations. Accordingly, particles forming tracks (as in Figure 3) were practically not formed, and, therefore, were not registered. The investigations showed that at a pressure of 30 Torr or less, it is easier to register particle tracks if an electrode with a small radius of curvature made of a heavy metal, such as tungsten, is used. It was found that each of the gases has its optimal pressure for observing the tracks. An image of several tracks from the high-voltage anode made of pointed tungsten wires 0.2 mm in diameter, tied into a bundle, is shown in Figure 4.

The image shows several tracks (see 1 and 2 in Figure 4a) formed by moving microparticles, starting from bright spots on the tungsten anode. Bright spots are also visible on the plane aluminum cathode, but there are no tracks under these conditions. This experiment shows that particles can appear even in the absence of a spark channel. The formation of bright spots on the electrode is sufficient for their initiation. Moreover, the electrode made of heavy metal turned out to be the most suitable for obtaining tracks with a large gap length. At pressures of different gases from ones to tens of Torr and with the tungsten electrode with a small radius of curvature, the intensity of the track glow decreased with distance from the electrode. In addition, in contrast to the conditions in Figure 3, they had a direct trajectory. The length of the tracks depended on the pressure and decreased with increasing pressure. At low pressures (less than 1 Torr), bright spots on were not formed the electrodes and no tracks were observed. The appearance of the tracks was affected by both the pressure and the type of gas. During a discharge in argon with the RADAN-220 generator, bright spots were formed on the tungsten electrode in a narrow pressure range compared to hydrogen; accordingly, this narrowed the range of conditions under which particle tracks were observed. The results shown in Figure 3 and Figure 4 show that pulsed discharges produce particles that, under certain conditions, can be observed from the tracks they leave behind. The particles appear due to the rapid heating of the electrodes at local points by the flowing discharge current.

Figure 4b shows a photograph in which the particles formed in the discharge can be seen. They were scraped off the side surface of the discharge chamber and placed on a microscope slide. Metal particles and their compounds with oxygen and nitrogen with a size of ~500 nm and less are non-uniformly distributed over the surface of the slide. In some regions, clusters of particles are visible (see region 2 in Figure 4b). The size of the particles depended on the discharge mode, the type of gas and its pressure, and the electrode material.

In addition to the formation of particles, spark discharges, and the appearance of bright spots on the electrodes, the evaporation and sputtering of the electrode material occurs. Moreover, when operating in a repetitively pulsed mode, the concentration of metal atoms increases [25]. Below are the results showing the effect of vapors of the electrode material on the discharge color, including at low pressures.

### 3.3. Effect of Electrode Material on the Color of Pulsed Diffuse Discharges

As is known, the color of a discharge plasma depends on the type of discharge and its operation mode, as well as on the composition of a gas mixture and pressure. In turn, the discharge formation process is determined by the amplitude and duration of a voltage pulse, the current density, the shape of the electrodes, and the interelectrode distance. In the pulse-periodic mode, the intensity of the discharge plasma glow increased. Under these conditions, this made it easier to photograph the discharges at low power consumption. Figure 5 shows photographs of air plasma glow during the discharge between two aluminum electrodes at different air pressures.

The discharge has a diffuse form at pressures of fractions of Torr and covers the entire volume of the discharge chamber, and its plasma glow has low-intensity radiation (Figure 5a,b). Its color is far from the bright red of the sprites and elves shown in Figure 1. However, it is closer to the color of the elves on the video from the ISS [15]. Due to the low density of the discharge current under these conditions, bright spots were not observed on the electrodes. Accordingly, the density of the metal vapors from the electrodes at air pressures of 1.5 and 3 Torr was low.

The color of the discharge in the gap became reddish when the gas pressure increased to 10 and 30 Torr, but it also did not correspond to the typical color of sprites (see Figure 1). So, as shown in Figure 5b,c, red spherical jets appeared near the bright spots on the electrodes. The color of these jets matches the color of the sprites. The differences in the color of the discharge plasma in the gap filled with air and near the bright spot on the cone-shaped aluminum electrode is also clearly seen in Figure 6a.

For the discharge in air with the cone-shaped aluminum electrode (Figure 6a), as in Figure 5c,d, the discharge color at the bright spot on the electrode changed and a red spherical jet appeared. When the gap width was decreased and the chamber was filled with argon, a bright spot also appeared on the plane electrode (Figure 6d). The color of the spherical jet near it was red, as was that near the cone-shaped electrode. Filling the discharge chamber with argon eliminates the influence of N_2_ (1P) radiation on the discharge color.

Experiments were also carried out with electrodes made of different metals. Photographs of the discharge plasma glow in the gap formed by a plane stainless steel anode and a cone-shaped aluminum anode are shown in Figure 6b,c. The use of aluminum electrodes changed the color of the discharge in argon, air, and nitrogen near the electrodes to red. On the surface of the cone-shaped electrode, bright spots surrounded by red halo can be seen. The number of bright spots depends on the discharge current density and gas pressure, as well as on the shape and polarity of the electrodes. As the interelectrode distance increases, the number of bright spots on the plane electrode usually decreases. The voltage pulse repetition rate also affects the number of bright spots. However, the color of the discharge plasma inside them mainly depends on the electrode material. Figure 6b,c demonstrates photographs of the discharge in the gap with a plane stainless steel electrode. The glow color of the plasma cloud (spherical jets) on the plane electrode in these figures is blue.

The use of copper electrodes resulted in an intense green color around the bright spots on the high-voltage electrode (Figure 7).

The lines of the copper atom dominate in the visible region of the emission spectrum (Figure 7b). An important role in the excitation of the above spectral transitions in copper atoms is played by the transfer of energy from the metastable level A^3^Σ_u_^+^ of molecular nitrogen [25]. A green glow was also observed during high-altitude discharges in the Earth’s atmosphere (ghosts in [16]), but the probability of their observation is negligible. In [16], these green-colored areas are located above the red sprites. They have a spherical shape and occur at an altitude of about 80 km.

It should also be noted that for the conditions in Figure 5, Figure 6 and Figure 7, as in [26], runaway electrons were registered behind a foil or mesh anode at pressures from ones to tens of Torr. These electrons pre-ionize air and other gases and contribute to the formation of diffuse discharges [21,27]. When TLEs appear in the Earth’s atmosphere, high-energy particles, including electrons, are also detected [23].

## 4. Discussion

Usually, the region emitting a certain color near bright spots on a copper, aluminum, or steel electrode (see Figure 5, Figure 6 and Figure 7) has a spherical shape. However, sometimes, it can also have the shape of a cylindrical jet. Figure 8 shows blue jets obtained in laboratory discharges. These jets correspond to ones in the video from the ISS [20].

Although the sizes of blue jets in the Earth’s atmosphere and in laboratory experiments differ by several orders of magnitude, their color and shape are similar. Atmospheric blue jets are usually in the form of a cylinder or a cone, from which thinner jets can propagate. The blue jets near the stainless electrode obtained at the discharge in the air (Figure 8b–c) had a similar shape.

The shape of red sprites is more complex. They are formed in two directions. Initially, one or several jets start from a plasma formation at an altitude of about 75 km [18,31] from the Earth’s surface to the ground. Next, the jet(s) or diffuse cloud propagate in the opposite direction. A similar picture was observed in our experiments (Figure 3). Double diffuse jets, starting from one region, propagated in opposite directions. To initiate a streamer (ionization wave) in an electric field, it is necessary to create plasma with a sufficiently high concentration of charged particles. This can be facilitated by, for example, the development of an electron avalanche or a preliminary discharge. An experiment on the registration of particle tracks showed that jets (streamers) could also be initiated at the ends of tracks. We assume that micrometeorites, entering the dense layers of the atmosphere, also create plasma trails with high concentrations of charged particles and initiate some TLEs. Primarily, they are ghosts, and, under certain conditions, they are red sprites.

A micrometeorite containing copper, sputtering and burning, can create a vapor cloud, as well as initiating an ionization wave (streamer), which, at high altitudes has a spherical shape. Discharges in copper vapor are green, while in nitrogen and oxygen there are no intense lines (bands) in this region of the spectrum. The experiments in [32] showed that the presence of copper vapor during an apokampic discharge increases the length of the plasma jet (streamer) and decreases the voltage at which it appears.

The red color of the positive column, similar to that of a sprite, was recorded in [33] during a continuous-glow discharge in air at a pressure of 1.2 Torr. However, at the cathode, the discharge plasma color had a blue tint. The sprites spread in both directions from the place of their initiation at an altitude of about 75 km; most of them were red along their entire length. Therefore, micrometeorites, and not only ionospheric/mesospheric inhomogeneities, can also contribute to the initiation of red sprites and influence their color [34].

## 5. Conclusions

This study showed that the vapors of the electrode material can significantly affect the color of the plasma glow of pulsed and pulse-periodic discharges. The largest change in color is observed near bright spots on the electrodes, primarily near electrodes with a small radius of curvature, as well as in the area where the spark channels are adjacent to the electrodes. Based on these results, it is possible to put forward a hypothesis about the influence of cosmic dust [13] on the color of parts of transient luminous events and their initiation.

## Figures and Tables

**Figure 1 nanomaterials-12-00652-f001:**
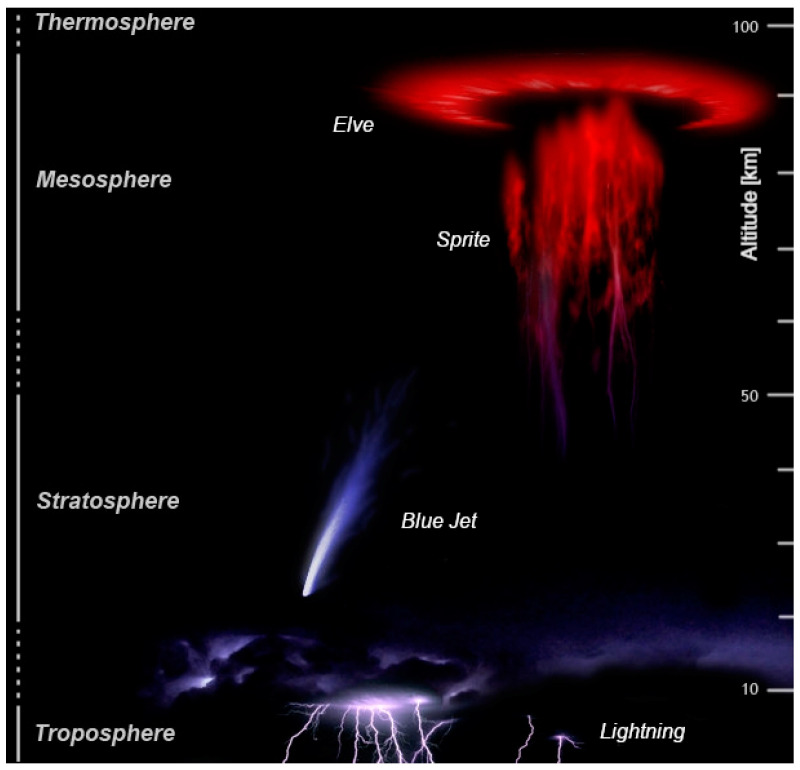
Different types of electrical phenomena in the atmosphere (TLE) [21]. Reproduced from https://en.wikipedia.org; Author: Abestrobi; accessed on 14 February 2022. The image is licensed under the Creative Commons Attribution-Share Alike 3.0 (CC BY-SA 3.0) Unported license).

**Figure 2 nanomaterials-12-00652-f002:**
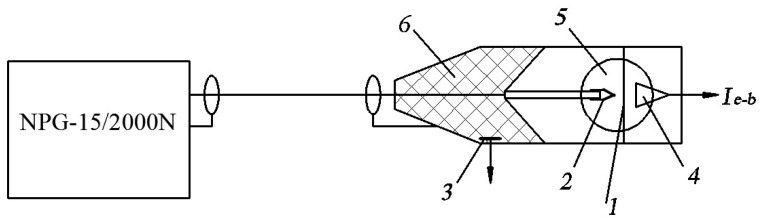
Sketch of the experimental setup. 1—plane anode, 2—cone-shaped cathode, 3—capacitive voltage divider, 4—collector, which was used to measure the runaway electron beam current, 5—discharge chamber with quartz side windows for capturing the discharge plasma glow, 6—insulator.

**Figure 3 nanomaterials-12-00652-f003:**
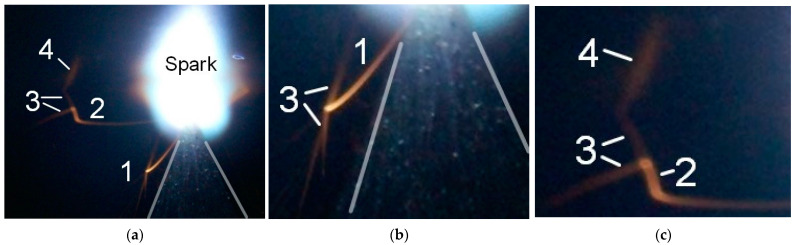
Integral images of the discharge in air at a pressure of 760 Torr with stainless steel electrodes. Single-pulse mode. NPG-18/3500N voltage pulse generator. The voltage pulse amplitude is 18 kV. The height of photograph (**a**) is 2 mm. The plane electrode is at the top and the cone-shaped electrode is at the bottom (outlines of the cone-shaped electrode in (**a**,**b**) are marked with white lines). The gap length is *d* = 1 mm. Zoomed images of the bottom 1 (**b**) and left 2 (**c**) tracks, which end with diffuse jets 3 and 4, are also shown.

**Figure 4 nanomaterials-12-00652-f004:**
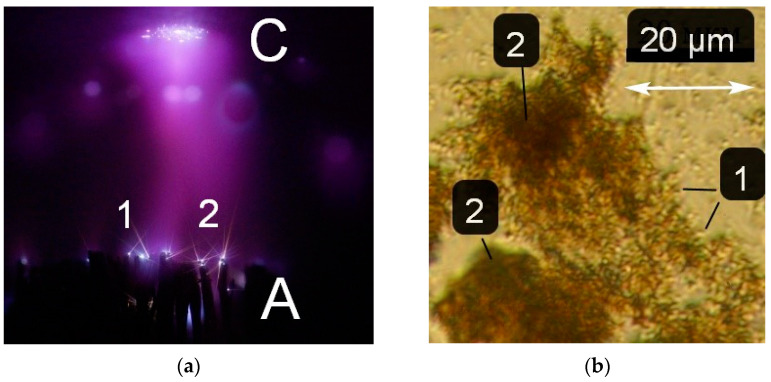
(**a**) Image of a discharge in hydrogen at ta pressure of 30 Torr with a tungsten wire anode (bottom) and a plane cathode made of aluminum (top), obtained during one voltage pulse (*U*_g_~120 kV, FWHM at matched load is 2 ns, rise time is 0.5 ns) from the RADAN-220 generator). C—cathode, A—anode. *d* = 5 mm. (**b**) Photograph of a part of the surface of a caprolon plate located at the sidewall of the discharge chamber opposite the discharge gap. Air pressure is 100 Torr, *d* = 2 mm. 1—nanoparticles with a transverse size of ~500 nm or less. 2—clusters of nanoparticles. NPG-15/2000N voltage pulse generator. *U*_g_ = 13 kV, *f* = 60 Hz. The cone-shaped electrode is made of copper, and the plane electrode is made of stainless steel.

**Figure 5 nanomaterials-12-00652-f005:**
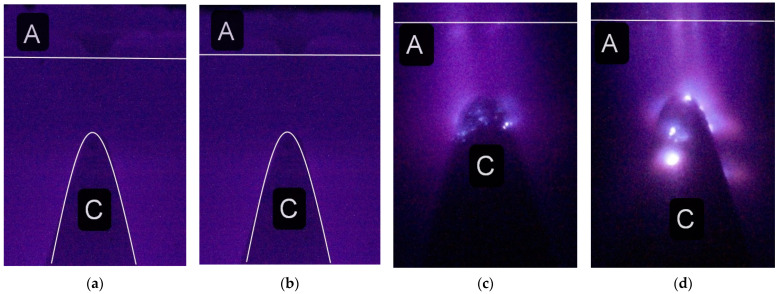
Photographs of the discharge in air at pressures of 1.5 (**a**), 3 (**b**), 10 (**c**), and 30 Torr (**d**) between two aluminum electrodes, obtained at *U*_g_ = 12 kV and *f* = 77 Hz. The plane electrode is located at the top and the cone-shaped electrode is located at the bottom. C—cathode, A—anode. *d* = 2 mm. Cathode (**a**,**b**) and anode (**a**–**d**) surfaces are marked with white lines. NPG-15/2000N voltage pulse generator.

**Figure 6 nanomaterials-12-00652-f006:**
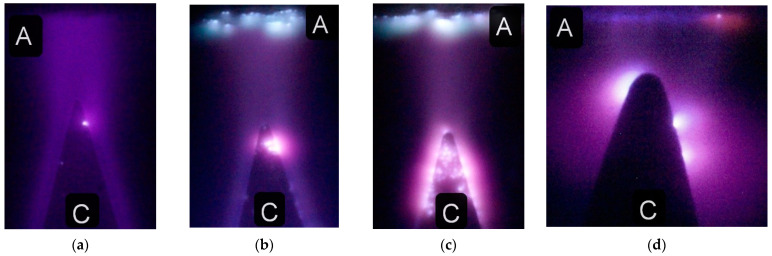
Images of the discharge in air at a pressure of 30 Torr (**a**) and in argon at pressures of 30 (**b**), 60 (**c**) and 10 Torr (**d**), obtained at *U*_g_ = 12 kV. *f* = 620 (**a**–**c**), and 77 Hz (**d**). In all photographs, the plane electrode made of aluminum (**a**,**d**) or stainless steel (**b**,**c**) is at the top, and the cone-shaped electrode made of aluminum is at the bottom. C—cathode, A—anode. *d* = 3 (**a**–**c**) and 2 mm (**d**). NPG-18/3500N voltage pulse generator.

**Figure 7 nanomaterials-12-00652-f007:**
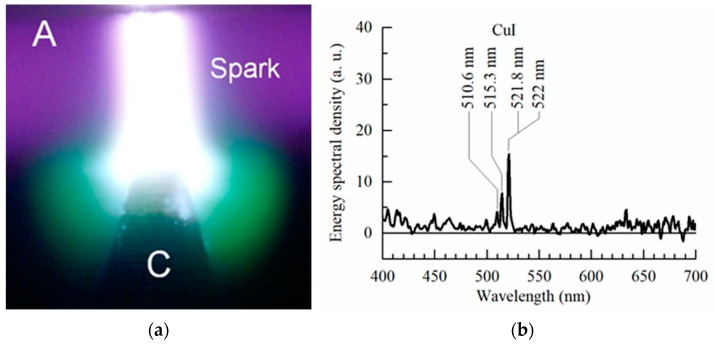
Photograph of the discharge in air at the pressure of 100 Torr (**a**) and the spectrum of the green discharge region near the cone-shaped copper cathode (**b**). The plane stainless steel electrode is located at the top. *f* = 60 Hz. C—cathode, A—anode. *d* = 2 mm. NPG-18/3500N voltage pulse generator. *U*_g_ = 13 kV.

**Figure 8 nanomaterials-12-00652-f008:**
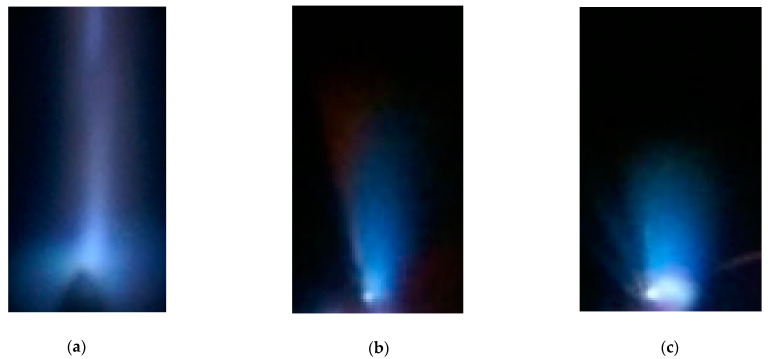
Photos of jets near the cone-shaped stainless steel cathode (**a**–**c**) at an air pressure of 760 Torr captured in one pulse. *d* = 2 (**a**) and 3 (**b**,**c**) mm. NPG-18/3500N voltage pulse generator.

## Data Availability

Data are contained within the paper.

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
