# Peer review of "Influence of Nanoparticles and Metal Vapors on the Color of Laboratory and Atmospheric Discharges"

_nanomaterials, 2022, doi:10.3390/nano12040652_

Round 1

Reviewer 1 Report

Review of the article

«Influence of nanoparticles and metal vapors on the color of la- 2 boratory and atmospheric discharges»

  Authors: Victor Tarasenko, Nikita Vinogradov, Dmitry Beloplotov, Alexander Burachenko, Mikhail Lomaev, and Dmitry Sorokin

The article is devoted to the study of the influence of nanoparticles and metal vapors on the color of laboratory and atmospheric plasma discharges.

The article may be of interest, but for researchers in the field of plasma physics and gas discharges. Of course, the article discusses the effect of particles, including possibly nanoparticles, on the discharge glow, but the article is not suitable for the journal's topics.

In addition, there are comments on the research methodology and interpretation of research on the article itself. Here are some of them:

1) Of course, metal vapors, nano- and micro-particles can affect the characteristics of the discharge. This is known from numerous studies of simpler and more understandable plasma objects such as glow discharge and arc. Similar studies and comparisons with known experimental data could be carried out on these objects. In this connection, the use of high-voltage pulse discharges is not justified. The authors draw an analogy with the formation of sprites, then, in my opinion, the study would be more suitable for another journal related, for example, atmospheric research.

2) The article is more descriptive in nature, there is no explanation of physical laws, but which are misleading readers. For instance:

Lines 168, 169

"An increase in the track brightness is apparently determined by an increase in the particle charge and size due to the evaporation of metal from the surface."

One can agree with the influence of the particle size on the track size, but how can an increase in the particle charge affect the track glow?

Lines 173-175

«Based on the brightness of the track glow, the plasma concentration at the particle surface increases with distance from the cathode.»

Again, it is not clear how the plasma density on the particle surface can affect the brightness of the track glow.

Such inaccuracies are found throughout the text.

In this regard, I would recommend that the authors revise the article (taking into account their colossal authority in the scientific community and the possibility of conducting more thorough experimental research) and send the article to a more specialized journal.

Author Response

Reply to Reviewer 1

The article: «Influence of nanoparticles and metal vapors on the color of laboratory and atmospheric discharges»

Authors: Victor Tarasenko, Nikita Vinogradov, Dmitry Beloplotov, Alexander Burachenko, Mikhail Lomaev, and Dmitry Sorokin

Submission Date: 14 December 2021

Date of this review 29 Dec 2021 14:04:26

Comments to the Authors

The article is devoted to the study of the influence of nanoparticles and metal vapors on the color of laboratory and atmospheric plasma discharges.

The article may be of interest, but for researchers in the field of plasma physics and gas discharges. Of course, the article discusses the effect of particles, including possibly nanoparticles, on the discharge glow, but the article is not suitable for the journal's topics.

Reply:

The manuscript has been revised. Several references was added. We hope that the results presented in the article will also be of interest to researchers who deal with micro and nanoparticles in traditional areas of nanomaterial physics [1-8]. Edits in the text are highlighted in either yellow or red font.

In addition, there are comments on the research methodology and interpretation of research on the article itself. Here are some of them:

1) Of course, metal vapors, nano- and micro-particles can affect the characteristics of the discharge. This is known from numerous studies of simpler and more understandable plasma objects such as glow discharge and arc. Similar studies and comparisons with known experimental data could be carried out on these objects. In this connection, the use of high-voltage pulse discharges is not justified. The authors draw an analogy with the formation of sprites, then, in my opinion, the study would be more suitable for another journal related, for example, atmospheric research.

Reply:

Our studies have shown that the effect of metal vapors on the color of the discharge plasma is much greater in the case of a pulsed breakdown in a nonuniform electric field. This made it possible to compare with the color of various discharges occurring in the upper atmosphere (Figure 1 in the text of the article). Even with well-studied discharges, it is very difficult to obtain such results. Thus, during an arc discharge, although there is a strong evaporation of the electrodes and the ejection of particles of various sizes, the voltage across the gap is low and the discharge in metal vapors outside the high-temperature arc channel is practically absent. Under these conditions, broadband Planck radiation from the arc channel plasma is observed. In a glow discharge, the sputtering of the electrodes has a low rate, and the emission spectra of the plasma of such discharges contain atomic and molecular transitions of the gases used. In the investigated mode, metal vapors are produced and excited by a pulsed discharge.

2) The article is more descriptive in nature, there is no explanation of physical laws, but which are misleading readers. For instance:

Lines 168, 169

"An increase in the track brightness is apparently determined by an increase in the particle charge and size due to the evaporation of metal from the surface."

One can agree with the influence of the particle size on the track size, but how can an increase in the particle charge affect the track glow?

Reply:

This judgment was removed from the text of the manuscript.

Lines 173-175

«Based on the brightness of the track glow, the plasma concentration at the particle surface increases with distance from the cathode.»

Again, it is not clear how the plasma density on the particle surface can affect the brightness of the track glow.

Reply:

Plasma concentration discussions have been removed from the text because no such measurements have been made. The nature of the glow of the particles in Figure 3 corresponds to the glow of a micrometeorite that burns down in the Earth's atmosphere [13] and [http://galaxy.astron.kharkov.ua/statti/meteor.htm]. The brightness of the micrometeorite (the particle) glow increases towards the track’s end. This cannot be explained by an increase in the particle velocity, since the particle stops. This occurs after the voltage pulse action. We believe that an increase in the radiation intensity of the particle is due to its heating during deceleration on gas particles. The text of the article has been revised.

Such inaccuracies are found throughout the text.

In this regard, I would recommend that the authors revise the article (taking into account their colossal authority in the scientific community and the possibility of conducting more thorough experimental research) and send the article to a more specialized journal.

Reply:

The text of the article was finalized and the English was improved.

We hope for a positive decision regarding the publication of this article in this journal.

From authors:

Victor F. Tarasenko and Dmitry A. Sorokin

Institute of High Current Electronics

E-mail: VFT@loi.hcei.tsc.ru

Reviewer 2 Report

  1. The introduction section could be improved. Some sentences are not well written or convey obvious ideas. Moreover, the aim of this work is to present an overview about the discharge behavior. The introduction would benefit from adding more content and detailed review overview to this research, such as particle lifting in electrostatic discharge , Turbulence effect, Mist-containing environment, Electrode materials.

  2. Please describe better your experimental system, such as ignition energy.
  3. The choice of the electrode material should be clearly explained in the present paper.
  4. Please explain the mechanism of different color arcs produced by electrodes of different materials.

Author Response

Reply to Reviewer 2

The article: «Influence of nanoparticles and metal vapors on the color of laboratory and atmospheric discharges»

  Authors: Victor Tarasenko, Nikita Vinogradov, Dmitry Beloplotov, Alexander Burachenko, Mikhail Lomaev, and Dmitry Sorokin

Submission Date: 14 December 2021

Date of this review: 19 Jan 2022 07:18:39

Comments and Suggestions for Authors

  1. The introduction section could be improved. Some sentences are not well written or convey obvious ideas. Moreover, the aim of this work is to present an overview about the discharge behavior. The introduction would benefit from adding more content and detailed review overview to this research, such as particle lifting in electrostatic discharge, Turbulence effect, Mist-containing environment, Electrode materials.

Reply:

The introduction of the manuscript and its text have been revised. In addition, new references have been added to the article. All changes in the text are highlighted in either yellow or red font. First of all, data describing the characteristics of the pulsed nanosecond discharge used were added and the influence of the electrode material was described in more detail. On the other hand, we should note that with a voltage pulse duration from ones to tens of nanoseconds, the experimental conditions differ significantly from the conditions of an electrostatic discharge and the creation of a foggy environment.

  1. Please describe better your experimental system, such as ignition energy.

Reply:

The experimental system was described in more detail, and various modes of discharge ignition were analyzed.

  1. The choice of the electrode material should be clearly explained in the present paper.

Reply:

In our preliminary studies, as well as in the papers of other authors, it was found that during pulsed discharges of short duration, the electrode material determines the composition of the vapors that evaporate from the electrodes and diffuse into the gap, including due to shock waves and turbulence. Since under these conditions diffuse discharges, at which the voltage across the gap remains high are formed, metal vapors are excited and ionized together with gas molecules in the discharge gap. This leads to the emission of radiation at various spectral transitions of metal atoms. Only a part of these transitions has a high radiation intensity in the region of interest to researchers. Also, to obtain a high intensity of radiation, transitions of atoms in metal vapors can be used, which are populated as a result of the efficient transfer of energy from excited gas molecules and atoms. Metals, the color of the emission of vapors of which, when excited in the plasma of nanosecond discharges, corresponds to the color of high-altitude atmospheric discharges, were chosen as the material of the electrodes.

  1. Please, explain the mechanism of different color arcs produced by electrodes of different materials.

Reply:

The color of the glow of the discharge plasma at electrodes made of various metals is associated with excitation certain energy levels of particles in the vapors of these metals. So, when using electrodes made of aluminum, steel and copper, we observed the glow of red, blue and green colors, respectively. The different colors of the discharge when changing the material of the electrodes are determined not by the spark or arc stages, but by bright spots on the electrodes, which are formed due to the explosive emission of electrons [Mesyats, G.A. Ecton mechanism of the vacuum arc cathode spot. IEEE transactions on plasma science, 1995, 23(6), pp. 879-883. (DOI: 10.1109/27.476469)]. These areas in the photographs have a bright white color (see, for example, the photographs in Figures 3, 6, 7). In spark or arc discharge, as well as in bright spots the electrodes are locally heated to a high temperature, which leads to the evaporation of the electrode material. High-temperature zones on the electrodes also supply micro- and nanoparticles into a discharge gap. However, emission of individual particles is determined by their temperature, it corresponds to the Planck radiation and is broadband. We note once again that in this work, to obtain metal vapors, as well as metal nano- and microparticles, a pulsed nanosecond discharge in a non-uniform electric field was used.

From authors:

Victor F. Tarasenko and Dmitry A. Sorokin

Institute of High Current Electronics

E-mail: VFT@loi.hcei.tsc.ru

Reviewer 3 Report

The paper provides interesting data concerning the colors of some specific kinds of plasma discharges in different gases and compares them to atmospheric discharges, considering the presence of metal vapors and nanoparticles coming from the electrodes.

The topic is timely and the results provided are interesting. However, most of the results provided consist in photographs of the discharges: while these are clear and descriptive, they provide a qualitative information only. A spectral analysis is given for one discharge only (discharge in air with a copper electrode): more quantitative results, such as a spectral analysis of the other discharges also, would be helpful for the comparison between the discharges produced in the lab and the ones observed in the atmosphere.

A point that should be clarified is the choice of the gases used for the discharges. One would expect a comparison between discharges that happen in gas mixtures of similar composition. Is there any relationship between the specific choice of gases made by the authors and the composition of the atmosphere at the altitudes where the atmospheric discharges take place?

A couple of requests of clarification about specific parts of the text are also provided in the following.

Page 5, line 168

The authors state: “An increase in the track brightness is apparently determined by an increase in the particle charge and size due to the evaporation of metal from the surface.”

The sense of this sentence is not completely clear to me. Where does the evaporation of the metal take place? It it took place on the particle surface, it should produce a reduction of its size, rather than an increase.

Page 6, line 218

The authors state: “Particles of metal and its compounds with oxygen and nitrogen with a size of 500 nm and less are nonuniformly distributed on the surface of the slide”. Some data should be provided to show the elemental composition of the particles.

Author Response

Reply to Reviewer 3

The article: «Influence of nanoparticles and metal vapors on the color of laboratory and atmospheric discharges»

  Authors: Victor Tarasenko, Nikita Vinogradov, Dmitry Beloplotov, Alexander Burachenko, Mikhail Lomaev, and Dmitry Sorokin

Submission Date: 14 December 2021

Date of this review: 01 Feb 2022

Comments and Suggestions for Authors

The paper provides interesting data concerning the colors of some specific kinds of plasma discharges in different gases and compares them to atmospheric discharges, considering the presence of metal vapors and nanoparticles coming from the electrodes.

The topic is timely and the results provided are interesting. However, most of the results provided consist in photographs of the discharges: while these are clear and descriptive, they provide a qualitative information only. A spectral analysis is given for one discharge only (discharge in air with a copper electrode): more quantitative results, such as a spectral analysis of the other discharges also, would be helpful for the comparison between the discharges produced in the lab and the ones observed in the atmosphere.

Reply:

The manuscript has been revised. Several references was added.

A point that should be clarified is the choice of the gases used for the discharges. One would expect a comparison between discharges that happen in gas mixtures of similar composition. Is there any relationship between the specific choice of gases made by the authors and the composition of the atmosphere at the altitudes where the atmospheric discharges take place?

Reply:

The main experiments were carried out with discharges in air, and the air pressure was chosen close to the pressures of high-altitude discharges. Other gases, such as argon, were chosen to better demonstrate the discoloration of the discharge.

A couple of requests of clarification about specific parts of the text are also provided in the following.

Page 5, line 168

The authors state: “An increase in the track brightness is apparently determined by an increase in the particle charge and size due to the evaporation of metal from the surface.”

The sense of this sentence is not completely clear to me. Where does the evaporation of the metal take place? It it took place on the particle surface, it should produce a reduction of its size, rather than an increase.

Reply:

This judgment was removed from the text of the manuscript. The manuscript has been revised.

Page 6, line 218

The authors state: “Particles of metal and its compounds with oxygen and nitrogen with a size of 500 nm and less are nonuniformly distributed on the surface of the slide”. Some data should be provided to show the elemental composition of the particles.

Reply:

Thanks for the advice. We plan to do this in our next work.

From authors:

Victor F. Tarasenko and Dmitry A. Sorokin

Institute of High Current Electronics

E-mail: VFT@loi.hcei.tsc.ru

Round 2

Reviewer 1 Report

The authors have revised the article. It can be accepted for publication.

Reviewer 2 Report

According to the comments of the first time, the author has made good supplement and modification.